# Shift-work sleep disorder among health care workers at public hospitals, the case of Sidama national regional state, Ethiopia: A multicenter cross-sectional study

**Adugnaw Adane**[1]*, **Mihret Getnet**[2], **Mekonnen Belete**[3], **Yigizie Yeshaw**[2], **Baye Dagnew**[2]

1 Department of Human Physiology, School of Medicine, College of Medicine and Health Sciences, Hawassa University, Hawassa, Ethiopia, 2 Department of Human Physiology, School of Medicine, College of Medicine and Health Sciences, University of Gondar, Gondar, Ethiopia, 3 Department of Human Physiology, School of Medicine, College of Medicine and Health Sciences, Wollo University, Dessie, Ethiopia

* adugnaw252@gmail.com

## Abstract

### Introduction

Shift-work disrupts circadian rhythm, resulting in disturbed sleep time and excessive sleepiness during the work shift. Little is known about shift-work sleep disorder among health care workers in Ethiopia. This study examined the magnitude and associated factors of shift-work sleep disorder among health care workers in Public Hospitals in Sidama National Regional State, Southern Ethiopia.

### Methods

An institution-based cross-sectional study was carried out on 398 health care workers selected using a systematic random sampling technique. A self-administered structured questionnaire consisting of insomnia, sleepiness scales and international classification of sleep disorder criteria items was employed. Epi data version 4.6 and Stata 14 were used for data entry and statistical analysis respectively. Binary logistic regression was fitted to determine associated factors and decision for the statistical significance was made at p<0.05 in the multivariable binary logistic regression.

### Results

Three hundred and ninety-eight health care workers (female = 53%) were included in the analysis with a response rate of 94.8%. The prevalence of shift-work sleep disorder was 33.67% (95% CI: 29.17%-38.45%). Being married (AOR = 1.88 (1.01–3.28)), three-shift (AOR = 1.078 (1.00–3.16)), ≥11 night shifts per month (AOR = 2.44 (1.36–4.38)), missing nap (AOR = 1.85 (1.04–3.30)), daily sleep time < 7hours (AOR = 1.88 (1.05–3.38)), khat chewing (AOR = 2.98 (1.27–8.09)), alcohol drinking (AOR = 2.6(1.45–4.92)), and cigarette smoking (AOR = 3.32 (1.35–8.14)) were significantly associated with shift-work sleep disorder.

**Data Availability Statement:** We described all the relevant information in the paper. The data set is sent to School of medicine for privacy of study

participants and the refined dataset can be obtained from the school of medicine (Dr. Mulugeta Ayalew; head of school of Medicine, email: yimermulugeta58@gmail.com) upon reasonable request.

**Funding:** The authors did not receive any specific funding for this work.

**Competing interests:** The authors have declared that no competing interests exist.

**Abbreviations:** BMI, Body mass index; ESS, Epworth sleepiness scale; ETB, Ethiopian birr; HCW, Health care worker; ICSD-2, International classification of sleep disorders, 2$^{nd}$ edition; ICSD-3, International classification of sleep disorders, 3$^{rd}$ edition; IQR, Interquartile range; ISI, Insomnia severity index; SD, Standard deviation; SWSD, Shift-work sleep disorder.

## Conclusion

This study showed a high prevalence of shift-work sleep disorder. Two shift schedule, napping, and reduction of substance use might reduce shift-work sleep disorder.

## Introduction

Shift-work requires a sleep-wake schedule that regularly conflicts with the natural, endogenous rhythm of sleep and wakefulness [1]. Work hours that result in non-standard sleep times can cause circadian misalignment, which impairs sleep, resulting in insomnia, severe sleep debt, and daytime sleepiness which result in shift-work sleep disorder [2].

Shift-work sleep disorder (SWSD) is a circadian rhythm disorder characterized by a chronic mismatch between a shift worker's sleep-wake schedule and his or her circadian clock [3]. Shift-work sleep disorder is determined by complaints of insomnia and/ or excessive sleepiness that occurs with work hours scheduled during the usual sleep period [4]. Based on the criteria for the diagnosis of SWSD in international classification of sleep disorder- 3$^{rd}$ edition (ICSD-3); SWSD is characterized by the complaint of insomnia and/or excessive sleepiness, symptoms associated with the shift work schedule lasting for $\geq$ 3 months, and circadian or sleep-time misalignment with a reduction of total sleep time and the disturbance is not explainable by other sleep disorders [5, 6].

The healthcare industry overall ranked third in the prevalence of insomnia and short sleep duration and in many countries healthcare workers make up the largest proportion of shift workers [7, 8]. It affects the health of workers negatively like short sleep, heart disease, metabolic syndrome, ulcers, cancer, obesity, and occupational accidents/injuries, fatigue and sleepiness, lack of psychosocial well-being, and job dissatisfaction are mentioned in many pieces of literature [9–11].

Globally, nearly 20% of the workforce is working in shifts that are outside standard work hours [11]. Shift-work sleep disorder was estimated to be 20%-30% among shift workers [2].

Approximately 10% of the USA workforce meets the criteria for SWSD [12]. A study in Italy showed SWSD was between 28% and 52% of HCWs while in the Netherlands 10–23% [13, 14]. The disease in Pakistan among nurses was 74.9% [15], Nigerian nurses 43.2% [16] and in Ethiopia it was 25.6% among nurses [17].

The factors that influence the development of sleep disorders in shift workers were mentioned in many pieces of literature like; age >50 years and female gender, being married, middle income and a length of service between 5 and 10 years or $\geq$10 years, night shift work, long working hours, short sleep (<6 h) and rotating shifts were reported to be associated shift work sleep disorders in studies Japan [18], Norway [19] Saudi Arabia [20], Ethiopia [17], Norway [21] and Nigeria [16].

In Ethiopia, shift work is increasing due to the necessity of providing 24hr medical services, which led hospitals to organize shift work schedules. Even though shift work has series of problems, there is a shortage of research findings conducted to show the magnitude and associated factors with SWSD in Health care workers in Ethiopia.

In Ethiopia, health care workers constitute most of the shift-working group but information regarding SWSD was not investigated remaining largely unnoticed. Furthermore, there is not much data available regarding the prevalence of shift work sleep disorder and associated factors. Therefore, this study was intended to show the prevalence and associated factors with SWSD among health care workers in public Hospitals in Sidama National Regional state,

Southern Ethiopia, 2021. We can use this finding to recommend health care administrators and professionals. Finally, increasing local and national resources for clinicians and scientists interested in this field for further study.

## Methods

### Study setting, design, and period

This institution-based cross-sectional study was conducted in Public Hospitals in Sidama National Regional state, Southern Ethiopia from 25 March 2021 to 10 May 2021. Sidama National regional state consisted of six town administrations and thirty districtswith an overall of 576 kebeles. The overall population of the region in 2020 has reached 5,493,516. There are fifteen Hospitals in the region, one comprehensive specialized hospital, three general, and Eleven District hospitals. Hawassa University comprehensive specialized hospital found in Hawassa city 273km away from Addis Ababa, Yirgalem General Hospital found in Dale Woreda in Sidama region, Adare general Hospital found in Hawassa city and Dore Bafano primary hospital found in Dore Bafano. These Hospitals provide preventive, curative, and diagnostic services to Sidama, Oromia, Gideon, and the surrounding populations

### Population

**Study population.**   All Health care workers involved in shift-work in the selected hospitals at the study area during the study period were included in the study.

### Eligibility criteria

All permanently employed healthcare workers who were working in the Hospitals for at least one year and involved in shift schedule for the last three months were included in the study.

### Sample size determination

The sample size was determined using single population proportion formula with the following assumptions: considering 50% prevalence of SWSD among health care workers of (no previous study among health care workers in the study area), 95% CI, 5% margin of error (d), and a 10% of non-response rate. The minimum calculated sample size was 384 and after adding non-response, the sample size became 423.

### Sampling technique

Systematic random sampling technique was used to select the participants from the list of all shift-working healthcare workers in the respective hospitals proportional to each after random lottery method selection of hospitals from Sidama National Regional state. The total number of health care workers, working in shift, in each hospital was 1488. Then determine K interval using total shift working Health care professionals (1,488/423), k = 3. Then the first sample was selected randomly and every third numbered health care worker was taken into the study in each selected hospital until the required number of study participants reached (Fig 1).

### Data collection and the questionnaire

Data collection was carried out using a structured self-administered questionnaire. A standardized SWSD diagnostic validated questionnaire by Barger (sensitivity = 0.74, specificity = 0.82) was adapted [22–24] and it was prepared in English language consisting of socio-demographic, sleep health, and behavioral factors. Insomnia Severity Index (ISI) was used to

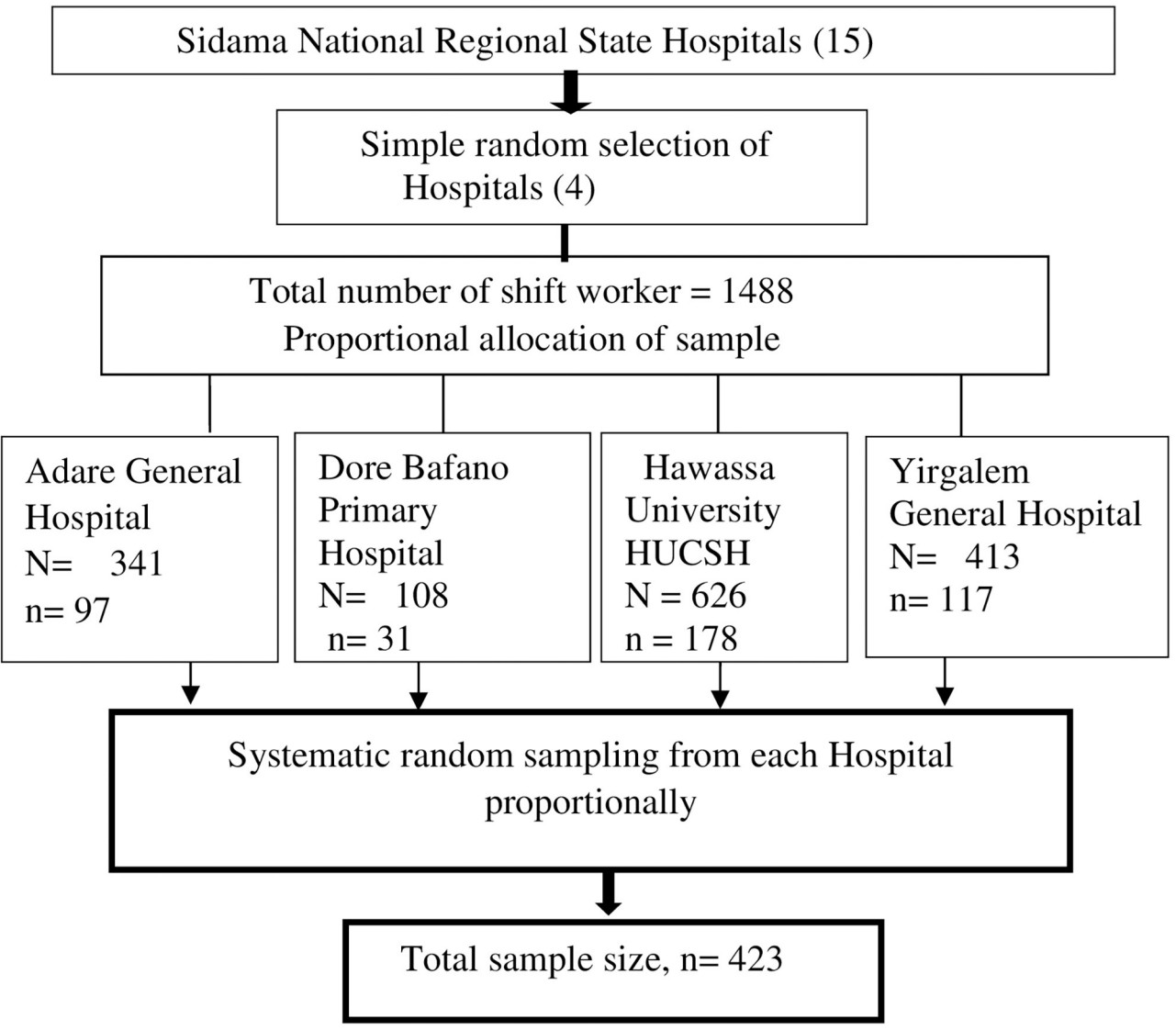

**Fig 1. Sampling procedure for assessment of shift-work sleep disorder in health professionals, Sidama, Ethiopia, 2021.**

determine insomnia [25], and Epworth sleepiness scale (ESS) was used to quantify excessive sleepiness which was validated in Ethiopia [26]. Four data collection facilitators and two supervisors (Nurses) were assigned from the selected Hospital. Orientation was given for facilitators about the purpose of the study and ethical issues to dispatch the questionnaire and explain the purpose of the study. After obtaining written consent from each participant, respondents fill the questionnaire.

## Measurement of variables

**Shift-work sleep disorder based on ICSD-3 criteria:** determined based on the participant's responding "yes" to the questions developed and used by many studies according to ICSD-3:

1. Do you experience difficulties with sleeping or excessive sleepiness? (Yes/no),

2. Is the sleep or sleepiness problem related to a work schedule where you have to work when you would normally sleep? (Yes/no),

3. Has this insomnia or sleepiness problem related to your work schedule persisted for at least three months? (Yes/no) [22, 27, 28].

**Shift-work sleep disorder (SWSD)**: Determined either after exploring those individuals who met ICSD-3 criteria or respondents who met ICSD-3 criteria and either ESS ≥11 or ISI ≥7 or both were considered to have SWSD.

**Insomnia Severity Index (ISI):** A self-administered insomnia severity index, with symptom-related questions based on the American Psychiatric Association Diagnostic and Statistical Manual of Mental Disorders-IV inclusion criteria for insomnia. The ISI has seven items scaled from 0 to 4. Participants were categorized as insomniacs if scoring a total score ≥ 7 [25, 29].

**Epworth Sleepiness Scale (ESS)**: The ESS constitutes eight items. Each item describes a specific situation for which respondents were asked to assess the likelihood of them falling asleep or dozing off on a scale ranging from 0 (would never doze off) to 3 (high chance of dozing off) [22, 26]. The ESS score cut-off "11" was used to declare excessive sleepiness.

**Two-shift workers**: Employs who work at noon as first shift or Night as the second shift with 12 hours rotation period.

**Three-shift workers**: An employee working third shift might start work around 11:00 p.m. or midnight and work until seven or eight in the morning with 8 hours rotation period.

## Data quality control

The data were collected using well-prepared English version of questionnaires after reviewing different literature and consultation of experienced experts in the subject area. To assure data quality, data collectors were trained for two days about how to use a pre designed form, how to handle study participants. A pretest was conducted on 24 Health care workers a week before the actual data collection. A constant monitoring was also part of this study and was framed as an integral part of the data collection processes. Every questionnaire was checked after data has been collected for an error and completeness and collected data was handled and stored properly until the analysis was done.

## Study variables

**Dependent Variable:** Shift-work sleep disorder (Yes/No)

**Independent variables: Sociodemographic variables** (Age, sex salary, profession, having children, educational level, working department)**, Behavioral and related factors (**Alcohol Use, khat chewing, coffee drinking, body mass index, Smoking cigarette, sleep medication),

**Shift-work and sleep health-related factors (**Working hours per week, presence of chronic disease, number of night shifts per month, daily sleep length, number of shifts, nap).

## Data management and statistical analysis

Data entry was performed using Epi data version 4.6 and then exported to Stata 14 for necessary visualization and recoding. The data exported was cleaned detecting the incomplete, incorrect and irrelevant records, and recoded, organized and transformed to comprehensible form. The data nature for continuous variables was also checked for normality with the Shapiro-Wilk test. Descriptive statistics were executed to summarize results according to data nature and tables and graphs were used to present the result. Binary logistic regression model was used to determine the associated factors with shift-work sleep disorder. Those variables

with a p-value $\leq$ 0.25 in the bivariable logistic regression were entered into a multivariable logistic regression model to be adjusted to other variables and crude and adjusted odds ratios with 95% CI were reported. The decision for the strength of association was made using the odds ratio and the statistical significance was decided at $p < 0.05$ in the multivariable binary logistic regression. The Hosmer-Lemeshow goodness of fit test was used for the model fitness.

### Validity and reliability of measurement tools

The tool used was validated in many studies Ethiopia and abroad [22, 26, 30]. We performed Cronbach's alpha coefficient to test the reliability of the Epworth sleepiness scale, Insomnia severity index, and ICSD-3 criteria items, which were used to assess SWSD, and we found a scale reliability coefficient of 0.76 for ESS, 0.77 for ISI, and 0.89 for ICSD-3 criteria items, which were acceptable reliability values.

### Ethical considerations

Ethical approval was obtained from the institutional review committee (IRC) of theSchool of medicine, college of medicine and health sciences, University of Gondar (ref. no.473/2021). Written informed consent was obtained from each study participant after explaining the significance of the study. Potential identifiers were not described in the questionnaire to ascertain confidentiality and the data collected was placed in a secured place.

## Results

### Socio-demographic characteristics of respondents

Three hundred and ninety-eight participants were included in the analysis and the response rate was 94.08%. The median age of the respondents was 31 (IQR = 7) years. Females were 53.3% of the study participants and 228 (57.29%) were married. Nearly 1 in 3 participants were bachelors. The Median income was 8017 (IQR = 3000) ranging from 3,830 to 20,000. One hundred and eighty-five (46.48%) and 74 (18.5%) were Nurses and Midwives, respectively. Moreover, 111 (27.5%) respondents were working at ward and 68 (17.09%) at the emergency departments. The median experience of the respondents was 7 years (IQR = 6) whereas 177 (44.47%) and 67(16.83%) respondents worked 5–9 years and $\geq$10 years respectively (Table 1).

### Sleep health and substance use behavior related characteristics

Among the total respondents, 245 (61.5%) work in two-shift, and 48.99% worked for over 10 nights per month. Half of the respondents (50.75%) reported a daily sleep time of fewer than 7 hours with a mean daily sleep time of 7.3 hours (SD = 1.4). One hundred five respondents (26.38%) missed a regular nap during night shift. More than half (53.5%) of participants complained excessive daytime sleepiness whereas 58% of them complained difficulty of initiating or maintaining sleep. One in five was currently drinking alcohol. Nearly 80% of respondents drank coffee daily whereas only 49 (12.3%) chew chat (Table 2).

### Prevalence of shift-work sleep disorder

The prevalence of sift-work sleep disorder was 33.67% (95%CI; 29.17%- 38.45%). Among participants who met the SWSD, criteria by ICSD-3, those who had insomnia (ISI $\geq$7) and/or Excessive sleepiness (ESS $\geq$11) were considered as having SWSD. From those who met ICSD-3 (n = 165), 134(81.2%) have SWSD. Similarly, 67.3% of those who respond to the insomnia severity index $\geq$7 (n = 199) and 76% of those who respond to the Epworth sleepiness scale $\geq$11 were found to have shift work sleep disorder (Fig 2).

**Table 1. Socio-demographic characteristics of health care workers working at public hospitals in Sidama national regional state, Southern Ethiopia, 2021 (N = 398).**

| Variables | Category | Frequency (%) | P-value |
|---|---|---|---|
| Sex | Male | 186(46.7) | |
| | Female | 212(53.3) | |
| Age(years) | ≤ 29 | 142(35.7) | <0.05 |
| | 30–34 | 144(36.2) | |
| | ≥ 35 | 112(28.1) | |
| Religion | Orthodox | 130(32.6) | |
| | Muslim | 65(16.3) | |
| | Protestant | 160(40.2) | |
| | Catholic | 24(6) | |
| | Others* | 19(4.7) | |
| Marital status | Single | 170(42.7) | <0.05 |
| | Ever married | 228(57.3) | |
| Educational level | Diploma | 86(21.62) | <0.01 |
| | Bachelor's degree | 273(68.59) | |
| | Masters & above | 39(9.8) | |
| Profession | Medical doctor | 40(10.05) | <0.01 |
| | Nursing | 185(46.48) | |
| | Midwifery | 74(18.59) | |
| | Laboratory | 53(13.32) | |
| | Pharmacy | 46(11.56)) | |
| Working Section | Emergency | 68(17.09) | <0.05 |
| | ICU/ NICU | 39(9.80) | |
| | Wards | 111(27.59) | |
| | Delivery | 43(10.80) | |
| | OR | 53(13.32) | |
| | laboratory | 47(11.81) | |
| | Pharmacy | 37(9.30) | |
| Monthly income (Birr) | ≤ 4999 | 16(4) | <0.01 |
| | 5000–9999 | 253(63.56) | |
| | 10000–14999 | 100(25.1) | |
| | ≥ 15000 | 29(7.3) | |
| Experience (years) | ≤4 | 154 | <0.05 |
| | 5–9 | 177(44.47) | |
| | ≥10 | 67(16.8) | |

* = Jehovah witness, Adventist, ICU = Intensive care unit, OR = Operation room, $X^2$ = Chi-square

## Factors associated with shift-work sleep disorder

Among sociodemographic, sleep and clinical and Behavioral variables entered into bivariable binary logistic regression, many of them were associated with SWSD. Age, marital status, educational status, profession, working department, experience, number of shifts, number of work nights per month, Nap, daily sleep time, work hours per week, alcohol consumption, cigarette smoking, Khat chewing, and body mass index were taken as candidate variables for multivariable logistic regression at p-value ≤ 0.25.

The output from multivariable binary logistic regression showed, frequency of shift, the number of night shifts per month, short sleep time (less than 7hrs), being married, no napping,

**Table 2. Sleep health and substance use behavior related characteristics of HCWs working at public hospitals in Sidama national regional state, Southern Ethiopia, 2021 (N = 398).**

| Variable | Category | Frequency (%) | $X^2$/t-test p value |
|---|---|---|---|
| Number of Shift | Two shifts | 245(61.55) | <0.05 |
|  | Three shifts | 153(38.44) |  |
| Night shifts per month | ≤ 10 nights | 203(51.01) | <0.01 |
|  | ≥ 11 nights | 195(48.99) |  |
| Regular Nap | No | 105(26.4) | <0.05 |
|  | Yes | 293(73.6) |  |
| Weekly work hrs. | ≤60 | 70(17.6) | <0.05 |
|  | 61–80 | 248(62.3) |  |
|  | ≥81 | 80(20.1) |  |
| Chronic disease | No | 382(96) | <0.01 |
|  | Yes | 16(4) |  |
| Daily Sleep time (hrs.) | <7 | 124(31) | <0.01 |
|  | ≥7 | 274(69) |  |
| Drink Alcohol | No | 311(78.14) | <0.01 |
|  | Yes | 87(21.86) |  |
| Drink Coffee | No | 83(20.8) | <0.05 |
|  | Yes | 315(79.2) |  |
| Coffee timing | Morning | 33(10.50 |  |
|  | Afternoon | 142(45.08) |  |
|  | Before bed | 40(12.7) |  |
|  | Regularly | 100(31.75) |  |
| Sleep medication | No | 376(94.5) | <0.01 |
|  | Yes | 22(5.5) |  |
| Regular Exercise | No | 253(63.5) | <0.05 |
|  | Yes | 145(36.5) |  |
| Khat chewing | No | 349(87.7) | <0.01 |
|  | Yes | 49(12.30 |  |
| BMI (kg/m$^2$ | <25 | 287(71.1) | <0.05 |
|  | ≥ 25 | 115(28.9) |  |

current alcohol drinking, cigarette smoking, and khat chewing were associated with shift-work sleep disorder after adjustment to confounding variables.

Married participants had 1.8 times higher odds of having SWSD (AOR.1.88, 95%CI, 1.03–3.42, p = 0.007). The odds of SWSD in participants with a three-shift work schedule was 1.78 (AOR: 1.78; 95%CI: 1.04–3.16, p = 0.04) times higher as compared to those with a two-shift work schedule. Health care workers with 11 or more-night shifts per month reported 2.4 times higher odds of developing SWSD (AOR: 2.44.;95% CI:1.36–4.38, p = 0.002).

The participants who reported no regular nap were 1.8 times more likely to have SWSD than their counterparts (AOR: 1.85; 95% CI: 1.04–3.30, p = 0.03). Those who slept shorter than 7 hours had 1.88 times higher odd of SWSD (AOR; 1.88; 95%CI; 1.05–3.38, p<0.05) compared to those who slept longer.

The odds of SWSD in current alcohol users was 2.6 times higher than those who did not drink alcohol (AOR: 2.65, 95%CI; 1.45–4.86, P = 0.01) whereas the odd of SWSD among khat chewers was 2.9 times higher (AOR; 2.98; 95%CI; 1.21–7.39, p<0.05). Moreover, the odd of SWSD among cigarette smokers were 3 times higher than non-smokers (AOR; 3.32, 95%CI; 1.35–8.14, p = 0.01) (Table 3).

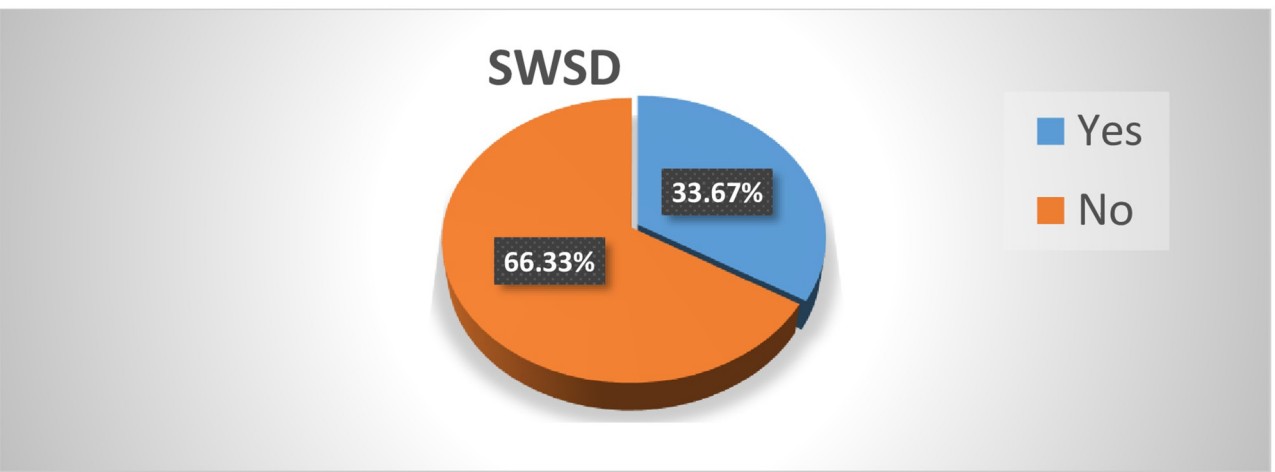

**Fig 2. Shift-work sleep disorder among health care workers working at public hospitals in Sidama national regional state, Southern Ethiopia 2021 (n = 398).** SWSD = Shift-work sleep disorder.

## Discussion

This study determined the prevalence of SWSD and its associated factors among HCWs using a survey questionnaire with SWSD diagnosis tools. The study found a significant number of HCWs had SWSD and showed, frequency of shift, the number of night shifts per month, being married, no nap during the night shift, short sleep time, cigarette smoking, use of alcohol and khat chewing were significantly associated with SWSD.

The prevalence of SWSD among HCWs was 33.67% (95% CI: 29%—38%). The result of the current study was similar to studies carried out in Norway 37.6% [19, 31], and another study in Norway 32.1% [21]. This finding is also consistent with the report by the American Academy of Sleep Medicine (10%-38%) [6]. This is also supported by studies in Italy (28–52%) [13]. However, the finding of the current study was lower than the prevalence of studies done in Pakistan (74.9%%) [15], India 39.9% [8], and Nigeria (43.2%), [16]. This result on the other side is higher than studies in Finland among health personnel (6%) [32] Japan among shift working nurses (24.4%) [18], Vietnam on HCWs (26%) [33], and Ethiopia among nurses (25.6%) [17]. The possible reasons for the difference might include study population; studies done in Japan, Pakistan, Nigeria, and Ethiopia were among nurses, which may not be generalized to other health professionals. Study setting where the health care and shift working system or habit may be different from the current study setting, furthermore, studies in Nigeria and Pakistan were a single institution. Another possible reason may be the assessment tool, where studies in the USA use objective tools like actinography and sleep log, which may rule out subjective complaints. The literature in Norway [19] uses ICSD-2 criteria to diagnose shift work sleep disorder while the current study uses ICSD-3 where using ICSD-2 may overestimate SWSD due to the inclusion of acute sleep disturbance complaints [5].

Married health care workers had 1.8 times higher odds of having SWSD compared to single. This result is in line with a study in China [20] and India [8]. This may be due to married shift-workers have additional social and family responsibilities that may affect sleep quality and duration in shift workers, which further interferes with the sleep cycle and expose them to SWSD. The timing of sleep is also socially regulated, and misalignment of sleep patterns from work and family obligations may contribute to SWSD [34].

**Table 3. Factors associated with SWSD among health care workers working at public hospitals in Sidama regional state, Southern Ethiopia, 2021 (n = 398).**

| Variable | Category | SWSD | | OR [95%CI] | |
|---|---|---|---|---|---|
| | | Yes (%) | No (%) | COR | AOR |
| Age (years) | ≤29 | 43(30.3) | 99(69.7) | 1.0 | 1 |
| | 30–34 | 49(34) | 95(66) | 1.18(0.72–1.96) | 0.73(0.39–1.37) |
| | ≥35 | 42(37.7) | 70(62.5) | 1.38(0.82–2.33) | 0.76(0.38–1.53) |
| Marital status | Single | 49 (28.8) | 121(71.2) | 1.0 | 1 |
| | Ever married | 85(38.3) | 145(61.7) | 1.46(0.95–2.25) | **1.88(1.01–3.28)** * |
| Educational status | Diploma | 15(17.4) | 71(82.6) | 1.0 | |
| | degree | 96(35.2) | 177(64.8) | 2.56(1.39–4.72) | 1.50(0.68–3.30) |
| | masters &above | 24(60) | 16(40) | 6.8(2.9–15.8) | 1.56(0.49–4.95) |
| Profession | Medical doctor | 24(60) | 16(40) | 3.1(1.28–7.49) | 1.77(0.39–7.97) |
| | Nursing | 54(29.2) | 131(70.8) | 0.85(0.42–1.70) | 0.71(0.22–2.34) |
| | Midwifery | 25(33.8) | 49(66.2) | 1.05(0.48–2.30) | 0.81(0.20–3.27) |
| | Laboratory | 16(30.2) | 37(69.8) | 0.89(0.38–2.09) | 0.43(0.04–4.20) |
| | Pharmacy | 15(32.6) | 31(67.4) | 1.0 | |
| Working department | Emergency | 16(23.5) | 52(76.5) | 0.80(0.34–1.88) | 0.50(0.06–4.11) |
| | ICU/ NICU | 21(53.84) | 18(46.14) | 3.05(1.24–7.48) | 1.05(0.10–10.30) |
| | Ward | 42(37.83) | 68(62.16) | 1.59(0.75–3.35) | 0.61(0.07–5.38) |
| | Delivery | 15(34.88) | 28(65.12) | 1.40(0.57–3.43) | 0.43(0.04–4.55) |
| | OR & recovery | 16(30.18) | 37(69.81) | 1.13(0.47–2.69) | 0.34(0.03–3.40) |
| | Pharmacy | 11(29.7) | 26(70.3) | 1.10(0.42–2.86) | 0.58(0.05–6.17) |
| | Laboratory | 13(35.13) | 34(64.86) | 1.0 | |
| Monthly Income(birr) | ≤4999 | 5(31.2) | 11(68.8) | 1.0 | |
| | 5000–9999 | 67(24.5) | 186(73.5) | 0.76(0.26–2.36) | 0.59(0.14–2.43) |
| | 10000–14999 | 43(43) | 57(57) | 1.65(0.53–5.13) | 1.01(0.22–4.68) |
| | ≥15000 | 19(65.5) | 10(34.5) | 4.18(1.13–15.4) | 1.47(0.25–8.35) |
| Experience (years) | ≤4 | 39(25.30) | 115(74.7) | 1.0 | |
| | 5–9 | 72(40.7) | 105(59.3) | 2.02(1.26–3.23) | 1.49(0.75–2.94) |
| | ≥10 | 23(34.3) | 44(65.7) | 1.54(0.83–2.87) | 0.72(0.26–1.97) |
| Number of shifts | Two shifts | 76(31) | 169(69) | 1.0 | |
| | Three shifts | 58(37.9) | 95(62.1) | **1.35(0.88–2.07)** | **1.78(1.00–3.16)** * |
| Night shift /month | ≤10 nights | 56(27.6) | 147(72.4) | 1.0 | |
| | ≥11 nights | 78(40) | 117(60) | **1.75(1.14–2.66)** | **2.44(1.36–4.38)** * |
| Sleep time (Hrs.) | <7 | 59(47.5) | 65(52.5) | **2.4(1.54–3.74)** | **1.88(1.05–3.38)** * |
| | ≥7 | 75(27.4) | 199(72.6) | 1.0 | |
| Work hours/week | ≤ 60 | 19(27.2) | 51(72.8) | 1.0 | |
| | 61–80 | 78(31.5) | 170(68.5) | 1.23(0,68–2.22) | 1.2(0.61–2.51) |
| | ≥81 | 37(46.2) | 43(53.8) | 2.30(1.16–4.58) | 1.4(0.61–3.36) |
| Napping | Yes | 89(30.4) | 204(69.6) | 1.0 | |
| | No | 45(42.8) | 60(57.2) | **1.71(1.08–2.72)** | **1.85(1.04–3.30)** * |
| Cigarette smoking | yes | 38(65.5) | 20(34.5) | **4.82(2.67–8.72)** | **3.32(1.35–8.14)** ** |
| | No | 96(28.2) | 244(71.8) | 1.0 | |
| Drinking coffee | yes | 115(36.5) | 200(63.5) | 1.9(1.10–3.39) | 1.2(0.64–2.46) |
| | No | 19(22.9) | 64(77.1) | 1.0 | |
| Drinking alcohol | Yes | 46(52.9) | 41(47.1) | **2.84(1.74–4.63)** | **2.65(1.45–4.86)** ** |
| | No | 88(28.3) | 223(71.7) | 1.0 | |
| Khat chewing | yes | 33(67.3) | 16(32.7) | **5.06(5.06–9.6)** | **2.98(1.21–7.39)** ** |
| | No | 101(29) | 248(71) | 1.0 | |

(*Continued*)

**Table 3.** (Continued)

| Variable | Category | SWSD | | OR [95%CI] | |
|---|---|---|---|---|---|
| | | Yes (%) | No (%) | COR | AOR |
| BMI (kg/m²) | <25 | 87(30.7) | 196(69.3) | 1.0 | |
| | ≥25 | 47(41) | 68(59) | 1.55(0.9–2.44) | 1.41(0.01–1.45) |

BMI (kg/m2) = Body mass index by kilograms per meter square,

* = significant with (p value <0.05),

** = significant at (p<0.01)

Goodness of fit test; p-value = 0.5803

Participants who had a three-shift work schedule were 1.78 times more likely to develop SWSD than two-shift workers. This was consistent with a report in Saudi Arabia revealing frequent rotating work schedules are a risk for SWSD [9], and Ethiopia [17]. The finding is contradictory to the result from a study in Japan [18] which states two-shift has more odds than three shifts to develop SWSD. The possible reason for the difference may be the difference in the study setting where the study was done; in our study setting, the shift is not purely 8hrs rotation where the nighttime is longer than daytime and evening, which increases the risk of shift work sleep disorder. Frequent rotation of working schedules and light/dark disturbance directly affects sleep by interfering with circadian shift [35]. Rotating shifts are often associated with shorter-than-normal and disrupted sleep periods resulting from circadian misalignment due to disruption of light/dark entrained sleep patterns. Rotating shifts can cause changes in the secretion of melatonin as a shift worker is awake at night and light at the workplace inhibit melatonin secretion, which is secreted at night and induce sleep [1].

Health care workers who work ≥11 nights per month were 2.4 times more likely to develop SWSD as compared to respondents working less than 11-night shifts per month. This finding is similar to study in Japan [18], India [8], Norway [19], Italy [36], and Ethiopia [17]. The reason might be frequent night shift restricts health care worker's opportunity for sleep and may spend many times for non-work activity between nights with daytime sleepiness, which may increase the tendency of SWSD. Night shifts disrupt the internal body clock as the entrained circadian rhythm urges to be awake during the daytime and sleep at night, the health personnel was awake at night, and night workers are likely to suffer from sleep loss, disrupted sleep, and fatigue which predisposes to SWSD [37].

Short daily sleep time (<7hours) had 1.8 times higher odds of SWSD. This is similar to the finding in Norway [21]. A shorter length of time leads to reduced sleep duration and may end up with circadian disruption. This may be because health care workers with rotating shift work schedules could be forced to stay awake for a long time at work in the night shift and result in short sleep duration, which disrupts the circadian rhythm, and homeostatic sleep requirements, which finally make them at risk for SWSD [12].

This study also reveals missing nap opportunities during the night shift has statistical significance with shift-work sleep disorder. Those who did not have naps during night shift were 1.8 times more likely to have SWSD than those who had regular napping. This is supported by studies in Japan [18], Italy [36], and the USA [38] which described having a regular nap during the night shift reduce SWSD. This may be because napping relieves sleep propensity accumulated during nighttime work, which in turn decreases worktime sleepiness and short sleep time, which are risks for SWSD [39]. Taking a 90 to 120 minutes nap during the night shift was reported to restore alertness and reduce sleep disorder risk [40].

Health care workers who drank alcohol were 2.6 times more likely to have SWSD compared to those who did not drink which is consistent with the findings in studies in Germany and Nigerian [16]. This is also supported by a study in Ethiopia which states alcohol consumption affects sleep quality, which might indirectly influence shift workers to develop SWSD. This might be due to the effect of alcohol on sleep patterns. It has been suggested that shift-workers may consume more alcohol than day workers as a sleep aid to compensate for sleep difficulties associated with work schedules [41]. Drinking alcohol near to sleep time was found to delay sleep time and affect the REM sleep pattern [42].

The odd of khat chewing was also found 2.9 times higher than HCW who do not chew khat. the substance self-administration pattern controls the timing of sleep and wake irrespective of the environmental light-dark cycle and additionally within sleep can alter the ultradian NREM-REM cycle [43]. Cathinone found in khat activates dopaminergic pathways involved in the regulation of sleep. Khat use is associated with poor sleep as well as other conditions implicated in precipitating sleep problems, i.e., anxiety, depression, and stress which further increase SWSD [44].

Cigarette smokers had a higher odd of developing SWSD than nonsmokers. This result is supported by the finding from Germany [45], and Italy [46] which state khat and cigarette smoking had a higher risk of sleep disorder. This may be due to the stimulant effect of cigarette, which inhibits sleep resulting in insomnia, and decreased sleep time, which finally result in sleep disturbance or short sleep duration, and shift workers who smoke cigarette are at higher risk of SWSD. This is also supported by a study in China which states smokers had a higher risk of insomnia and thus sleep disorder [47]. This may be because consumption of stimulant (nicotine) acutely disrupts sleep patterns which increases risk of SWSD in shift workers. Nicotine is a stimulant and makes it harder to fall asleep and to stay asleep by activating dopaminergic neurons in sleep regulatory centers. Cigarettes should ideally be avoided altogether, and certainly for at least 2 hours before bed [44].

Even though Older age group, work experience >10 years, drinking coffee, ICU working department, which were found to be associated with SWSD in many literatures [18–20, 45], these were not associated with SWSD in this finding. In these literatures the proportion of old age participants were higher than this study which was very small and re-categorized into >35, which makes it insignificant in the adjusted analysis. Coffee drinking is very common and cultural in the study setting and majority of participants drink coffee, which make it insignificant. Having work experience >10 years and ICU working department were stated in some literatures to be associated with SWSD [46, 48, 49].

The number of participants who had >10 years' experience and those who are working at ICU in the current study are small in number which were 16% and 9.8%, respectively which might make it insignificant in the multivariable regression analysis.

Those who have BMI >25 and work hour > 40 hrs/week were also stated to be associated with SWSD in some literatures [13, 50, 51]. But this study founded that there is significant association between SWSD and BMI. Even though BMI is associated with SWSD in bi-variable regression model it does not show independent association in adjusted model. This may be due to the difference in sociodemographic and economic differences which may greatly affect BMI and obesity which directly affects sleep quality and adequacy.

This study indicates the magnitude of SWSD to be high and shift work and substance use related variables were associated with SWSD. Therefore, it implies the need for prompt action on the health system in shift work and shifts workers to achieve the health need of the people and keep the health of health care workers.

This finding might be used as an input for the general shift-work system regulation, management, and research for the health sector and researchers on this area.

### Limitations of the study

The use of subjective tools to measure SWSD was the limitation of the study where using sleep logs or actinography brings better findings. The study design, which cannot show cause-effect relation since exposure and outcome were measured at a point in time. There might be also recall bias for questions that require memorizing events in the past.

## Conclusion

The findings of this study indicate the prevalence of shift-work sleep disorder is high among health care workers at public hospitals in Sidama National Regional State. Being married, frequency of shift, number of night shifts per month, missing naps, short sleep time, smoking cigarettes, khat chewing, and drinking alcohol were significantly associated with SWSD.

This alarming finding requires prompt action to protect the health of the health care providers. These findings also provided evidence that health care workers employed in shifts particularly throughout nights are prone to shift-work sleep disorder. Since this cross-sectional study lacks identification of causality, further studies on shift work and the burden of SWSD on HCWs with objective tools and longitudinal design are required to make better conclusions and recommendations.

## Acknowledgments

We would like to acknowledge the University of Gondar, Hawassa University, facilitators, and study participants.

## Author Contributions

**Conceptualization:** Adugnaw Adane, Mihret Getnet, Baye Dagnew.

**Data curation:** Adugnaw Adane.

**Formal analysis:** Adugnaw Adane, Mihret Getnet, Mekonnen Belete, Yigizie Yeshaw, Baye Dagnew.

**Investigation:** Adugnaw Adane, Mihret Getnet, Mekonnen Belete, Yigizie Yeshaw, Baye Dagnew.

**Methodology:** Adugnaw Adane, Mihret Getnet, Mekonnen Belete, Yigizie Yeshaw, Baye Dagnew.

**Project administration:** Adugnaw Adane.

**Resources:** Adugnaw Adane.

**Software:** Adugnaw Adane, Mihret Getnet, Mekonnen Belete, Yigizie Yeshaw, Baye Dagnew.

**Supervision:** Adugnaw Adane, Mihret Getnet, Baye Dagnew.

**Validation:** Adugnaw Adane, Mihret Getnet, Mekonnen Belete, Yigizie Yeshaw, Baye Dagnew.

**Visualization:** Adugnaw Adane, Mihret Getnet, Mekonnen Belete, Yigizie Yeshaw, Baye Dagnew.

**Writing – original draft:** Adugnaw Adane.

**Writing – review & editing:** Adugnaw Adane, Mihret Getnet, Mekonnen Belete, Yigizie Yeshaw, Baye Dagnew.

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
