## [Decision Letter · Decision Letter 0]

27 Dec 2021

PONE-D-21-37990Shift-work sleep disorder among health care workers at public Hospitals, the case of Sidama national regional state, Ethiopia: Multicenter cross-sectional studyPLOS ONE

Dear Dr. ADANE,

Thank you for submitting your manuscript to PLOS ONE. After careful consideration, we feel that it has merit but does not fully meet PLOS ONE’s publication criteria as it currently stands. Therefore, we invite you to submit a revised version of the manuscript that addresses the points raised during the review process.

Please revise the manuscript according to both Reviewers' suggestions.Please submit your revised manuscript by Feb 10 2022 11:59PM. If you will need more time than this to complete your revisions, please reply to this message or contact the journal office at plosone@plos.org. Please include the following items when submitting your revised manuscript:A rebuttal letter that responds to each point raised by the academic editor and reviewer(s). You should upload this letter as a separate file labeled 'Response to Reviewers'.A marked-up copy of your manuscript that highlights changes made to the original version. You should upload this as a separate file labeled 'Revised Manuscript with Track Changes'.An unmarked version of your revised paper without tracked changes. You should upload this as a separate file labeled 'Manuscript'.

We look forward to receiving your revised manuscript.

Kind regards,

Claudio Liguori

Academic Editor

PLOS ONE

Journal Requirements:

"All the authors decleared that no compiting interests exist."

5. Please include a copy of Table 1 which you refer to in your text on page 10.

Reviewers' comments:

Reviewer's Responses to Questions

**Comments to the Author**

1. Is the manuscript technically sound, and do the data support the conclusions?

Reviewer #1: Yes

Reviewer #2: Yes

2. Has the statistical analysis been performed appropriately and rigorously? 

Reviewer #1: Yes

Reviewer #2: Yes

3. Have the authors made all data underlying the findings in their manuscript fully available?

Reviewer #1: No

Reviewer #2: Yes

4. Is the manuscript presented in an intelligible fashion and written in standard English?

Reviewer #1: Yes

Reviewer #2: Yes

5. Review Comments to the Author

Reviewer #1: Abstract

Pg 2 Ln 29: formatting issue (capitalise and remove comma)

Ln 31: were used

Ln 35/36: no need for space between number and %

No further comments on formatting will be made (unless absolutely necessary). The authors are requested to please run the manuscript through an English language professional or through an appropriate software to make all the necessary changes.

Methods: Data Management and Statistical Analysis

Pg 9 Ln 174: why was such a high p-value chosen?

Results:

Socio-demographic characteristics of respondents

Pg 10 Ln 196: “first-degree holders” is not a common term – use “bachelors” or “x number of years of education/schooling”

Sleep health and substance use behavior related characteristics

Pg 10 Ln 206: what is meant by two-shift/ three-shift? This should be explained in the methodology

Pg 11 Ln 212: grammar and spelling (here and elsewhere, drunk = drank)

References

Please check formatting and spellings

Reviewer #2: Reviewer Name: Kalkidan Haile

Institution and Country: Debre Markos Comprehensive specialized hospital, Ethiopia

Please leave my comments for the authors below

Methods:

Dealing with an increasingly debated topic concerning shift work and its associated consequences on workers' health is essential for the research communities. The research is well designed and analysis is correct in a big sample. However the following comments need to be corrected.

• Under measurement of variable ( line 136-146) please specify the measurement

• Limitations are almost similar with previous studies. Your action make different by minimizing the limitation is poor.

Results

• I suggest discussing factors that have no association in the current studies but have association in previous researches with possible reasons why not have association in the current study.

Statistical Analysis

-Please describe the procedures for editing and cleaning the data.

-Please describe how predictor variables were entered into the logit models (e.g., entry, forward, backward, hierarchical).

-Were any potential confounding variables controlled for? If yes, please indicate.

6. PLOS authors have the option to publish the peer review history of their article (what does this mean?). If published, this will include your full peer review and any attached files.

Reviewer #1: No

Reviewer #2: **Yes: **Kalkidan Haile

---

## [Author Response · Author response to Decision Letter 0]

1 Feb 2022

We are pleased to thank both the reviewers and editor for the valuable comments. We have seen the comments and revised the manuscript according to the comments given and we also uploaded the answers for each comments given. we have resubmitted the revised manuscript following all the journal guidelines.

---

## [Decision Letter · Decision Letter 1]

18 Mar 2022

PONE-D-21-37990R1Shift-work sleep disorder among health care workers at public Hospitals, the case of Sidama national regional state, Ethiopia: A multicenter cross-sectional studyPLOS ONE

Dear Dr. ADANE,

Thank you for submitting your manuscript to PLOS ONE. After careful consideration, we feel that it has merit but does not fully meet PLOS ONE’s publication criteria as it currently stands. Therefore, we invite you to submit a revised version of the manuscript that addresses the points raised during the review process.Please control if the statistics can be improved according to the Reviewer suggestion.

We look forward to receiving your revised manuscript.

Kind regards,

Claudio Liguori

Academic Editor

PLOS ONE

Journal Requirements:

Reviewers' comments:

Reviewer's Responses to Questions

**Comments to the Author**

1. If the authors have adequately addressed your comments raised in a previous round of review and you feel that this manuscript is now acceptable for publication, you may indicate that here to bypass the “Comments to the Author” section, enter your conflict of interest statement in the “Confidential to Editor” section, and submit your "Accept" recommendation.

Reviewer #2: (No Response)

2. Is the manuscript technically sound, and do the data support the conclusions?

Reviewer #2: Yes

3. Has the statistical analysis been performed appropriately and rigorously? 

Reviewer #2: Yes

4. Have the authors made all data underlying the findings in their manuscript fully available?

Reviewer #2: Yes

5. Is the manuscript presented in an intelligible fashion and written in standard English?

Reviewer #2: Yes

6. Review Comments to the Author

Reviewer #2: Good to see the paper with such substantial change. However, the following things should be taken in to consideration.

1. I was commented to describe how predictor variables were entered into the logit models (e.g., entry, forward, backward, hierarchical). However, please ignore it because it does not apply to STATA.

2. On the discussion part, I still recommend you to discuss factors that don’t have an association in the current study but had associations in other previous studies.

3. Line 435-437, incorrectly cited. The correct citation is Haile KK, Asnakew S, Waja T, Kerbih HB. Shift work sleep disorders and associated factors among nurses at federal government hospitals in Ethiopia: a cross-sectional study. BMJ Open. 2019 Aug 27;9(8):e029802

7. PLOS authors have the option to publish the peer review history of their article (what does this mean?). If published, this will include your full peer review and any attached files.

Reviewer #2: No

---

## [Author Response · Author response to Decision Letter 1]

18 May 2022

We would like to thank you for your constructive and valuable comments and suggestions that would improve the manuscript. We have addressed all comments and suggestions one by one and hence revised the manuscript thoroughly.

---

## [Decision Letter · Decision Letter 2]

12 Jun 2022

Shift-work sleep disorder among health care workers at public Hospitals, the case of Sidama national regional state, Ethiopia: A multicenter cross-sectional study

PONE-D-21-37990R2

Dear Dr. ADANE,

We’re pleased to inform you that your manuscript has been judged scientifically suitable for publication and will be formally accepted for publication once it meets all outstanding technical requirements.

Kind regards,

Claudio Liguori

Academic Editor

PLOS ONE

Additional Editor Comments (optional):

Reviewers' comments:

Reviewer's Responses to Questions

**Comments to the Author**

1. If the authors have adequately addressed your comments raised in a previous round of review and you feel that this manuscript is now acceptable for publication, you may indicate that here to bypass the “Comments to the Author” section, enter your conflict of interest statement in the “Confidential to Editor” section, and submit your "Accept" recommendation.

Reviewer #2: All comments have been addressed

2. Is the manuscript technically sound, and do the data support the conclusions?

Reviewer #2: Yes

3. Has the statistical analysis been performed appropriately and rigorously? 

Reviewer #2: Yes

4. Have the authors made all data underlying the findings in their manuscript fully available?

Reviewer #2: Yes

5. Is the manuscript presented in an intelligible fashion and written in standard English?

Reviewer #2: Yes

6. Review Comments to the Author

Reviewer #2: (No Response)

7. PLOS authors have the option to publish the peer review history of their article (what does this mean?). If published, this will include your full peer review and any attached files.

Reviewer #2: No

---

## [Editor Report · Acceptance letter]

29 Jun 2022

PONE-D-21-37990R2 

Shift-work sleep disorder among health care workers at public Hospitals, the case of Sidama national regional state, Ethiopia: A multicenter cross-sectional study 

Dear Dr. Adane:

I'm pleased to inform you that your manuscript has been deemed suitable for publication in PLOS ONE. Congratulations! Your manuscript is now with our production department. 

Kind regards, 

on behalf of

Dr. Claudio Liguori 

Academic Editor

PLOS ONE